# HUMAN-MACHINE COOPERATION FOR SEMANTIC FEATURE LISTING

**Kushin Mukherjee, Siddharth Suresh & Timothy Rogers**
Department of Psychology
University of Wisconsin-Madison
Madison, WI 53706, USA
`{ kmukherjee2, siddharth.suresh, ttrogers}@wisc.edu`

## ABSTRACT

Semantic feature norms — lists of features that concepts do and do not possess — have played a central role in characterizing human conceptual knowledge, but require extensive human labor. Large language models (LLMs) offer a novel avenue for the automatic generation of such feature lists, but are prone to significant error. Here, we present a new method for combining a learned model of human lexical-semantics from limited data with LLM-generated data to efficiently generate high-quality feature norms.

**Introduction.** A central goal in cognitive science is to characterize human knowledge of concepts and their properties. Many have used human-generated feature lists as norms for establishing the structural relationship between concepts in the human mind (McRae et al., 2005; Devereux et al., 2014; De Deyne et al., 2008; Buchanan et al., 2019), but this requires extensive human labor. Large language models (LLMs) have recently shown impressive capabilities when generating properties of objects (Hansen & Hebart, 2022) or answering questions(Ouyang et al., 2022; Brown et al., 2020; Hoffmann et al., 2022; Chowdhery et al., 2022; Wei et al., 2021) and thus suggest an avenue for more efficient characterization of human knowledge structures, but even state-of-the-art models can routinely fail on many common-sense questions of fact. GTP3-davinci, for instance, will deny that alligators are green, while asserting that they can be used to suck dust up from surfaces. Thus, human effort can generate high-quality norms, but with prohibitive costs, while LLMs can produce norms with little human effort, but with considerably less accuracy. This paper considers whether human and machine effort can combine to efficiently estimate high-quality semantic feature vectors. Our approach leverages two key observations. First, the concept-by-feature matrices collected in norming experiments are typically high in dimension but low in rank—thus a low-dimension matrix decomposition based on a representative subset of concepts can be used to estimate how subsets of features covary with one another across concepts. Second, open-source LLMs optimized for question-answering, such as Google's FLAN-T5 XXL (Wei et al., 2021), can be used to generate "guesses" about which concepts possess which features. While these guesses may not be fully accurate, they can be be used to infer where a *new* concept might reside within the low-dimension similarity space computed from human-generated features. The predicted coordinates of the new item can then be re-composed to the original human feature space to make inferences about its features. This idea suggests the following procedure:

1. Collect human feature norms for $n$ concepts representative of a domain and assemble in concept-by-feature matrix $H$.
2. Compute a singular value decomposition of $H$, retaining the matrix of the first $d$ singular values times the first $d$ right singular vectors.
3. For a new concept, use FLAN-T5 XXL to answer questions about which properties are 'true' for the concept to create a binary vector of LLM-generated features, $h$.
4. Fit a logistic regression model to predict the new concept's $d$ left singular vector coordinates using the $d$ right singular vectors as predictors and $h$ as the target.
5. Multiply out the predicted left singular vectors through the rest of the SVD to get predictions about features of the new item.

Refer to Figure 1 A. for a visual schematic of the procedure. We evaluated this approach using a well-known pre-existing dataset of semantic feature norms De Deyne et al. (2008).

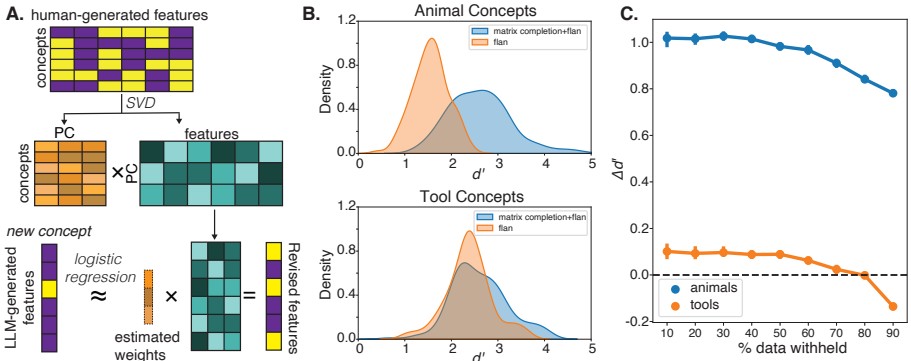

Figure 1: **(A.)** Our procedure for estimating features for new concepts. A human feature-listing matrix is decomposed using (SVD) into low-rank matrices of principal components (right) and weights to linearly combine them to estimate the features for each item (left). For a new concept, logistic regression models estimate weights to map the principal components to LLM-generated features for the concept. Multiplying the estimated weights with the principal components improves the alignment between humans and machines. **(B.)** $d'$ distributions for raw LLM features and LLM features generated from the process in (A). **(C.)** The difference in $d'$ values using raw LLM features and the method in (A) with different amounts of the original dataset withheld.

**Method.** The human norm dataset included 129 animal concepts evaluated on 764 features, and 166 tools evaluated on 1262 features. Each feature corresponded to a property generated by human participants when asked to 'list all the features they knew to be true of each concept'. For each domain separately, 4 human raters evaluated whether a given feature was generally true of a given concept (e.g. "Yes or no, is the property *is green* true of the concept *alligator*"). We threshold the values in these matrices such that only features that all 4 raters agreed to be true were treated as valid 'trues'. This gave us a set of ground truth binary human-generated feature matrices. Next, we used the XXL variant of FLAN-T5, a 11B parameter language model accessed through the huggingface transformers library (Wolf et al., 2019), to perform the same feature verification task for every cell in both matrices (see Appendix). We then treated human/machine feature comparisons as a signal detection problem in which the human norms provided the ground truth (signal present/absent) against which the machine responses were evaluated as hits, misses, false-alarms, or correct rejections. From these data we computed $d'$ as the measure of agreement between human and machine matrices. This metric was computed separately for the animal and tool datasets using just the FLAN-verified feature vectors. The central question was whether the matrix completion-based data-augmentation approach described above improves the match between machine and human feature vectors.

**Results.** Figure 1 A illustrates the general workflow. To test if our matrix completion technique improved alignment between human and machine features, we iteratively held-out one concept at a time and computed the SVD and regression models on the remaining concepts. We then estimated the features for the held-out concept and computed its $d'$ relative to ground-truth human features. We found that our matrix completion technique led to higher $d'$ values relative to using the raw features generated by the LLM for both animal ($t = 15.86$, $p < .001$) and tool ($t = 3.79$, $p < .001$) concepts (Figure 1B.).

For this method to be efficient, it should require little data to fit the SVD model. To estimate how much data would be needed for our matrix completion method to show meaningful gains, we held out 10% - 90% of the original human data when fitting the SVD model and estimated the differences in $d'$ when using our method vs. raw LLM feature lists. We found statistically significant improvements using our method while holding out up to 70% of the original data, with much greater improvements for animals than tools (Figure 1 C.).

**Conclusion.** We evaluated a new method for combining human effort and LLM behavior to more efficiently generate semantic feature norms. Such norms have been critical for understanding the nature and structure of human conceptual knowledge, but prior efforts have required extensive human labor. Our method yields higher-quality vectors than those produced by LLMs alone, with

potentially huge savings in human effort–opening the door for mapping a much broader set of concepts in future.

URM STATEMENT

The authors acknowledge that at least one key author of this work meets the URM criteria of ICLR 2023 Tiny Papers Track.

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

## A  APPENDIX

### A.1  FLAN-T5 FEATURE VERIFICATION

We used a standardized prompt for querying FLAN-T5 about whether a given feature was applicable for each concept in the animal and tool concept sets. Table 1 shows an example of a prompt to query whether the feature 'has two eyes' is valid for the concept 'Dolhpin'. We prefaced the question with the same example of a positive and negative example for the concept 'book'. We repeated this procedure for every concept x feature combination in the norm dataset collected by De Deyne et al. (2008)

---

Q: Is the property [is_female] true for the concept [book]?
A: False
Q: Is the property [can_be_digital] true for the concept [book]
A: True
In one word True/False, answer the following question
Q: Is the property [has_two_eyes] true for Dolphins?
A: <mask>

---

Table 1: Example Prompt.

## A.2  $d'$ AS AN ALIGNMENT METRIC

Since both the ground-truth human feature matrices and LLM-generated feature matrices had binary entries, the problem of human-machine comparison can be described in terms of signal detection theory. That is, if we treat the human-matrix as being the source of 'signal', the predictions of 1s and 0s in the machine matrix can be — (1) Hits if the cell in the matrix was 1 for both the human and machine matrix, (2) Misses if the cell in the machine matrix is 0 and 1 in the human matrix, (3) False alarms if the cell in the machine matrix is 1 and 0 in the human matrix, and (4) Correct rejections if the cell in both the machine and human matrices is 0. The number of hits, misses, false alarms, and correct rejections can be tallied to compute hit rate (HR) and false-alarm rate (FAR) as follows -

$$HR = \frac{hits}{hits + misses} \tag{1}$$

$$FAR = \frac{false\ alarms}{false\ alarms + correct\ rejections} \tag{2}$$

Finally, $d'$ is computed as

$$d' = z(H) - z(FAR), \tag{3}$$

where $z$ is the inverse of the cumulative distribution function (CDF) of the standard normal distribution $\mathcal{N}(0, 1)$. A higher $d'$ indicates a greater degree of alignment between human and machine features.

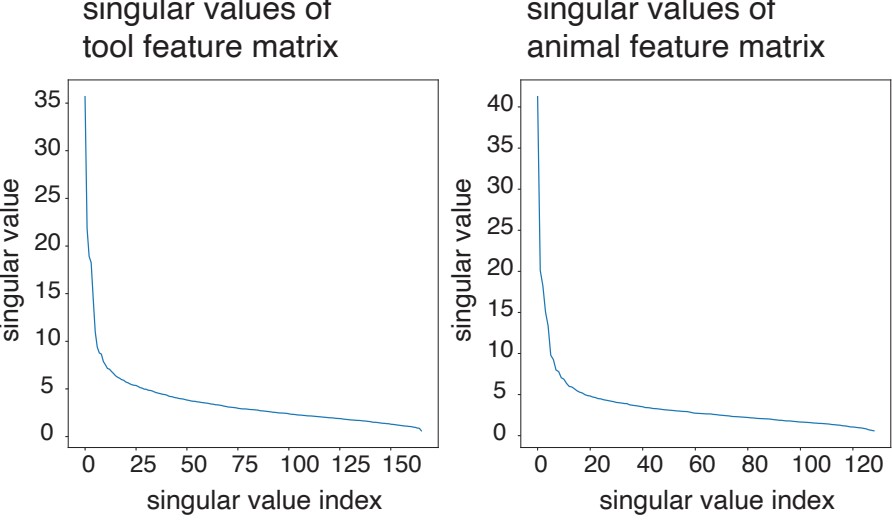

Figure 2: Profile of singular values for both human feature matrices visualized as scree plots. As can be observed, these matrices were low-rank, thus when fitting our feature verification model, we used a rank-10 decomposition when computing the SVD. We did find minor variations in performance as a function of changing the rank of the decomposition, but overall as long as the rank was around 10, which corresponds to the elbow of the scree plots, the reconstructed values were accurate.

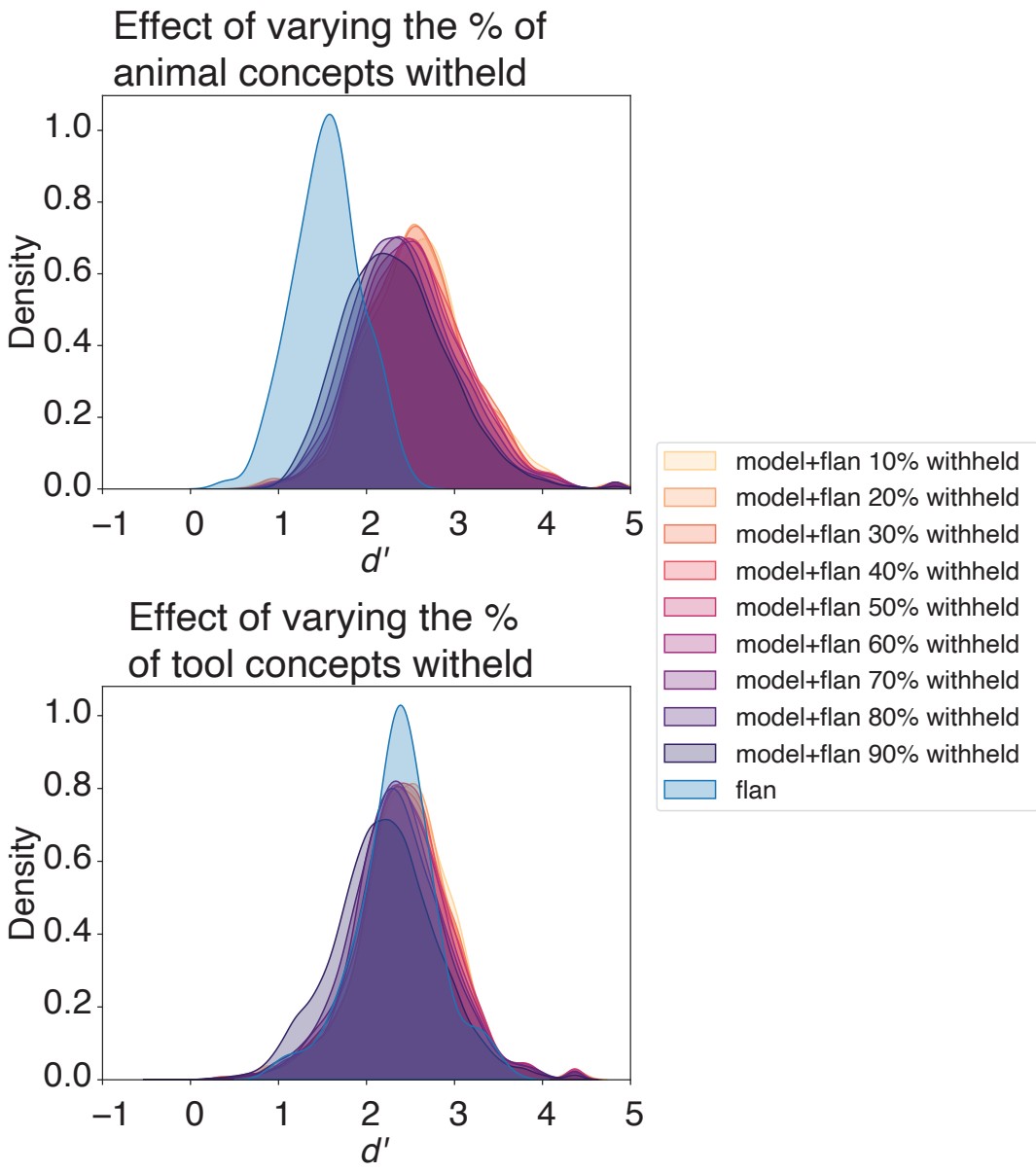

Figure 3: $d'$ distributions for different holdout percentages. As the amount of data used to fit the matrix-completion model is reduced, the more the model-based distribution overlaps with the pure LLM distribution (shown in blue). While greater gains are seen for the animal concepts, FLAN is already quite good at estimating the structure of tool concepts.

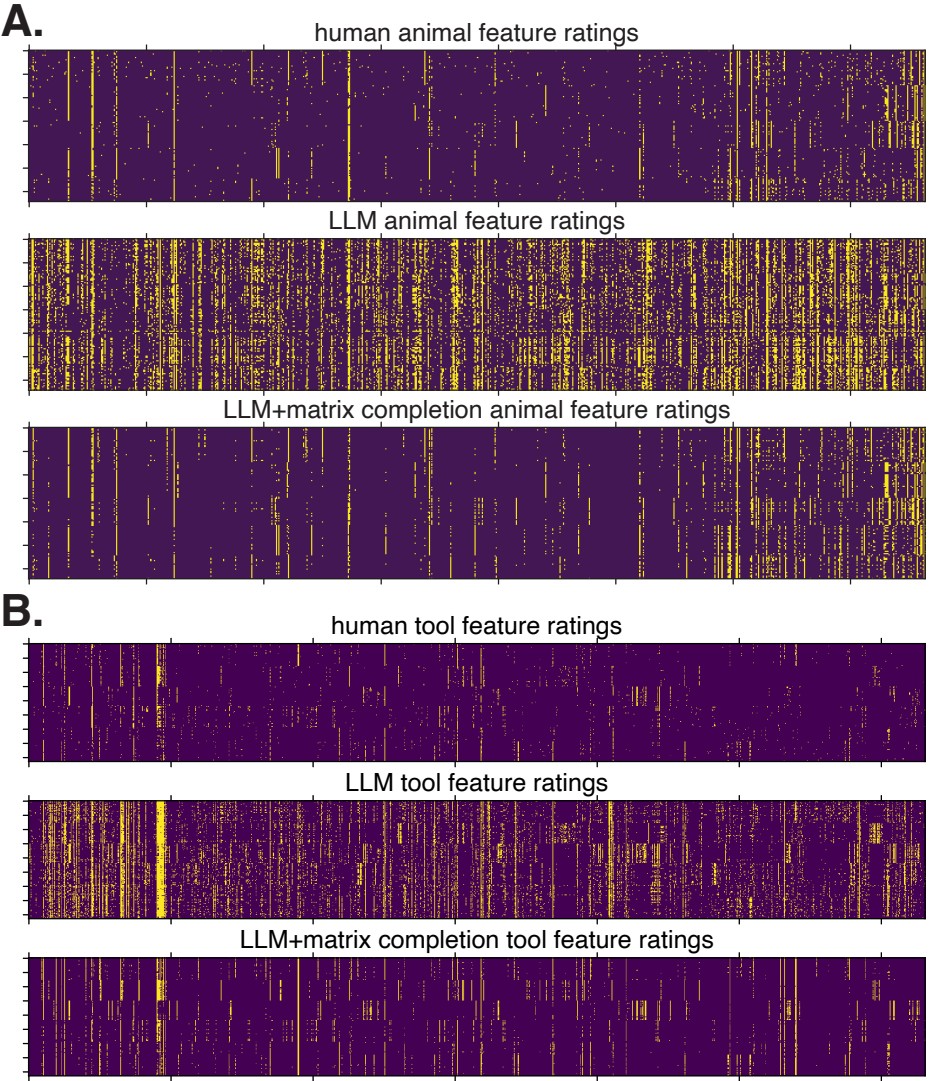

Figure 4: Human feature-ratings, FLAN-based LLM generated feature-ratings, and features generated using our matrix-completion model using a leave-one-out approach for (A) animal concepts and (B) tool concepts . Yellow cells correspond to 1s and purple cells correspond to 0s. The LLM tends to provide many false positives as is indicated by the greater proportion of yellow cells. Our method helps to 'clean' the LLM generated matrix to bring it into closer alignment with human ratings.

