# OpenReview forum: "Human-machine cooperation for semantic feature listing"
_ICLR.cc/2023/TinyPapers — Submitted to Tiny Papers @ ICLR 2023_

### Official Review · Reviewer_Y92m · 2023-03-31

**Confidence:** 4

**Summary Of Contributions:**

The authors propose a technique to combine the powerful but less accurate LLM generated norms with the restrictive but highly accurate human lexical-semantics to efficiently generate high-quality feature norms. The authors tested this combination on 129 animals and 166 tools concepts.

**Rating:**

High Potential (HP): a submission which meets the reviewing criteria and has potential to make an impact on the field

**Strengths And Weaknesses:**

- Clarity: The paper was easy to follow and relevant related works were cited.

- Correctness:
    - There is a huge difference in performance between concepts. What does -ve delta d mean in figure 1.c?
    - It is unclear whether the proposed techniques work only for one type of concept. Delta d with animals concept is much higher.

- Reproducibility: No code is provided but I see enough details in Appendix to reproduce the analysis.

- Follows basic requirements: yes


**Suggested Changes:**

- Specify what is deltaD and what values it can take and which way is better.
- I would also like to see some examples of the LLMs-generated concepts using the proposed technique make sense.

---

### Official Review · Reviewer_6wFZ · 2023-04-01

**Confidence:** 3

**Summary Of Contributions:**

The paper proposes a method to improve on raw LLM-generated features by doing matrix completion. Results show that the proposed method mostly improve over the raw features.

**Rating:**

Clear, Correct, and Reproducible (CCR): a submission which meets the reviewing criteria

**Strengths And Weaknesses:**

Strengths:

* This is a very timely idea. Since human annotation is often the most costly step of a learning process, using LLMs in innovative ways to help with the process is an interesting problem.
* The SVD-based approach seems to be robust in the sense that even with high levels of data withholding, it gives reasonable improvement over the raw LLM-generated features.

Improvement scopes:

* It's not reported how the number of features vs number of concepts affect this result. For example, if #features < #concepts, how well would the logistic regression, and in turn, the full process perform?
* I understand why the current SVD + logistic regression based method is a great start. But it's well worth thinking of improved methods for getting the revised features. In particular, maybe use of somewhat deeper models and techniques for self-supervised learning to get embeddings. Specially since the output of the logistic regression seems akin to embeddings already.


**Suggested Changes:**

The paper seems well-written and self-contained for the most part, as a tiny paper should be. The only change I would suggest is to add some definition/intuition about the measure of agreement, $d'$, in the main paper.

---

### Official Review · Reviewer_xUqV · 2023-04-02

**Confidence:** 4

**Summary Of Contributions:**

This paper presents a method for combining a human lexical-semantic model with LLM-generations to generate high-quality semantic feature norms

**Rating:**

Great Start (GS): a submission which meets some of the reviewing criteria but has room for improvement

**Strengths And Weaknesses:**

### Strengths ###
- Follows basic requirements: The submission meets formatting requirements and the page limit.

### Weaknesses ###
- Clarity: The methods mentioned in this paper are similar to retrofitting embeddings, but instead of retrofitting to a word sense or task, the LLM embeddings/feature space is retrofitted to a human-generated feature. It would be useful to mention prior retrofitting literature on this and show additional baselines to compare this method to prior techniques.
- Correctness: The claims and conclusions are partially justified by the findings. They can be reinforced if a comparison with other baselines is added.


**Suggested Changes:**

- It would be useful to mention prior retrofitting literature on this and show additional baselines to compare this method to prior techniques.

---

### Author Response · Authors · 2023-06-01
**Opt-in for archival**

We wish to opt in for archival for this submission.

---

### Meta-Review · Area_Chair_nTP1 · 2023-04-05

**Recommendation:** Invite to archive
**Confidence:** 4

**Metareview:**

This work studies an interesting problem of combining human effort and LLM behavior. Overall, this paper is timely and the idea is technically sound. The presentation is good. The reviewers have provided various suggestions to make the experiments and analysis more solid.








**Summary:**

This work proposes a new method for combining human effort and LLM behavior to more efficiently generate semantic feature norm. By testing the combination on 129 animals and 166 tools concepts, the alignment between human and machine features is improved with the proposed approach. The reviewers noted that this work is timely, technically sound, and easy to follow. The suggestions are mainly about analysis.

**Comments And Feedback To The Authors:**

There are several suggestions to improve the paper.

(i) Discuss prior retrofitting literature on this and show additional baselines to compare this method to prior techniques.

(ii) Show how the number of features vs number of concepts affect this result.

(iii) Improved methods for getting the revised features, e.g., use of deeper models and techniques for self-supervised learning to get embeddings.

(iv) Specify what is deltaD and what values it can take and which way is better.

(v) Show examples of the LLMs-generated concepts using the proposed technique.




**Reason For Not Giving A Higher Recommendation:**

There are concerns about clarity in the experimental design and reports. More baselines and improved features are encouraged to add.

**Reason For Not Giving A Lower Recommendation:**

This paper studies an interesting problem that will be of interest to the community. The paper is self-contained for the most part as a tiny paper.

---

> ### Author Response · Authors · 2023-06-01
> **Response to AC**
>
> We thank the Area chair for their comments and address some of the larger points below
>
> > (i) Discuss prior retrofitting literature on this and show additional baselines to compare this method to prior techniques.
>
> Our work is centered more around using the output of GPT3-level performant LLMs to estimate human conceptual structure as opposed to finding projections for the embeddings underlying models like FLAN to better adhere to our needs.
> We did review some of the retrofitting literature and did find some papers that were interesting to consider (https://arxiv.org/pdf/1909.09700.pdf & https://arxiv.org/pdf/2005.00524.pdf) but in the end, we felt that this was not directly relevant to the work at hand and would actually confuse the reader and hence chose to not include any references to this particular literature.
>
> > (ii) Show how the number of features vs number of concepts affect this result.
>
> We report experiments where we systematically vary the # of concepts (Fig 1. C. and appendix Fig 3.). Our general result shows gains even with a large percentage of concepts withheld. We do not report experiments varying the # of features as the goal of this approach is to uncover the entire set of features for given concepts, assuming some high quality data (with all the features included). In general it is not clear what a principled way to withhold features would be in this paradigm, but we do emphasize the sensitivity of our results to varying number of concepts now.
>
> > (iv) Specify what is deltaD and what values it can take and which way is better.
>
> We include a section on this in the Appendix due to space constraints.
>
> > (v) Show examples of the LLMs-generated concepts using the proposed technique.
>
> This suggestion is quite warranted. We are working on a detailed notebook, which will be in an associated GitHub repo to include specific examples.

---

### Decision · Program_Chairs · 2023-04-08

Invite to archive